# Efficacy and immunogenicity of rKVAC85B in a BCG prime-boost regimen against H37Rv and HN878 *Mycobacterium tuberculosis* strains

Eunkyung Shin[1], Jin-Seung Yun[1,2], Young-Ran Lee[3], Hye-Ran Cha[4], Soo-Min Kim[1], Sung-Jae Shin[5], Sang-Won Lee[6], Gyung Tae Chung[1], Dokeun Kim[1], Jung Sik Yoo[1], Jong-Seok Kim[7]*, Hye-Sook Jeong[1]*

**1** National Institute of Infectious Diseases, National Institute of Health, Korea Disease Control and Prevention Agency, CheongJu, Chungbuk, Republic of Korea, **2** Department of Biochemistry, College of Life Science and Biotechnology, Yonsei University, Seoul, Republic of Korea, **3** Bio-Convergence R&D Division, Korea Institute of Ceramic Engineering and Technology, CheongJu, Chungbuk, Republic of Korea, **4** Department of Microbiology and Immunology, Institute for Immunology and Immunological Diseases, Graduate School of Medical Science, Yonsei University College of Medicine, Seoul, Republic of Korea, **5** Department of Microbiology, College of Medicine, Yonsei University, Seoul, Republic of Korea, **6** Department of Data Science, Korea Disease Control and Prevention Agency, CheongJu, Chungbuk, Republic of Korea, **7** Myunggok Medical Research Institute, College of Medicine, Konyang University, Daejeon, Republic of Korea

* jeongnih@korea.kr (H-SJ); jskim7488@konyang.ac.kr (J-SK)

## Abstract

*Mycobacterium tuberculosis* infection accounted for 1.3 million deaths worldwide in 2022. Bacillus Calmette-Guérin (BCG) is the only licensed vaccine against tuberculosis (TB); however, it has limited protective efficacy in adults. In this study, we constructed a recombinant vaccinia virus expressing Ag85B from *M. tuberculosis* using a novel attenuated vaccinia virus (KVAC103). We then analyzed the immunogenicity of prime-boost inoculation strategies using recombinant KVAC103 expressing Ag85B (rKVAC85B) compared to BCG. In both rKVAC85B prime-boost and BCG prime-rKVAC85B boost inoculation regimens, rKVAC85B induced the generation of specific immunoglobulin G (IgG) and secretion of interferon-γ by immune cells. *In vitro* analysis of *Mycobacterium* growth inhibition revealed a comparable immune-mediated pattern of outcomes. Furthermore, bacterial loads in the lungs were significantly lower in mice inoculated with the BCG prime-rKVAC85B boost than in the BCG-only group following a rechallenge infection with both H37Rv and HN878 strains of *M. tuberculosis*. These findings collectively suggest that KVAC103, incorporated into a viral vector, is a promising candidate for the development of a novel TB vaccine platform that is effective against multiple *M. tuberculosis* strains, including H37Rv and HN878, and that rKVAC85B effectively stimulates immune responses against *M. tuberculosis* infection.

**Data availability statement:** All relevant data are within the manuscript and its Supporting Information files.

**Funding:** This research received financial support through grant No. 6637-300 from the National Institute of Health, encompassing both intramural and external research (2016-NG48002-00, 2019-NI085-00, and 2016-ER4202-00). The funders had no role in study design, data collection and analysis, decision to publish, or preparation of the manuscript.

**Competing interests:** The authors have declared that no competing interests exist.

## Introduction

Tuberculosis (TB), caused by *Mycobacterium tuberculosis*, is the second leading cause of death from single infectious agents, with 7.5 million newly diagnosed cases and 1.3 million deaths in 2022. TB also seriously threatens public health worldwide, similar to human immunodeficiency virus infection and malaria [1]. BCG, a live attenuated vaccine developed using *Mycobacterium bovis*, is commonly used to vaccinate newborns, effectively preventing severe incidence and disease transmission in childhood and adolescence [2–4]. However, the duration of protective efficacy is unstable in adults and adolescents, and the efficacy of BCG varies by age, area, and ethnicity. Additionally, the rates of infection and death due to TB have rapidly increased after the coronavirus disease 2019 pandemic, particularly in low-to-middle-income countries. Therefore, developing novel vaccines, either BCG replacements or BCG boosters, is urgently needed. Currently, 14 vaccine candidates are in different stages of clinical trials and are expected to prevent *M. tuberculosis* infections and disease development [5,6]. Most vaccines in clinical trials comprise protein subunits combined with adjuvants; viral vectors based on modified vaccinia virus, adenovirus, or influenza virus; or mRNA vaccines.

In the realm of TB prophylaxis, the Bacillus Calmette-Guérin (BCG) vaccine has been the predominant intervention in the last century, effectively reducing TB incidence in individuals from infancy to young adulthood. However, the efficacy of the BCG vaccine varies and decreases in the adult population, creating a major gap in TB control, particularly in the absence of alternative vaccines [1,2,7]. Current research is heavily focused on developing innovative TB vaccines that either supplement or enhance the immunity provided by BCG. Research in this regard has focused on various advanced immunological approaches, such as heterologous prime-boost strategies that utilize protein subunits, RNA-based vaccines, and viral-vectored vaccines [7–13]. These methods aim to amplify the initial immune response triggered by BCG, overcoming its limitations and offering extended protection in adulthood.

According to the TB Vaccine Initiative's TB vaccine development pipeline, clinical trials are underway to identify viral vector-based vaccine candidates, including those derived from MVA and chimpanzee adenovirus vectors [14]. These trials are exploring innovative vaccination regimens that involve monologous or heterologous prime-boost strategies, utilizing these viral vectors in various combinations to enhance their immunogenicity and efficacy against TB [5]. Using viral vectors such as MVA, adenovirus, and parainfluenza virus 5 effectively enhances the immunity elicited by the BCG vaccine [11]. This enhancement is primarily attributed to the induction of robust CD4+ and CD8+ T-cell mediated immune responses, which include antigen-specific multifunctional T-cells and IFN-γ-secreting cells, crucial for the efficacy of TB vaccines [12,15–18].

Viral vector vaccines, including MVA 85A, a modified vaccinia Ankara vaccine, have consistently remained in the TB vaccine pipeline. The attenuated vaccinia virus has been used as a vaccine platform as it can deliver and express foreign or large gene(s) and induce strong CD4+ and CD8+ cellular immunity. Hence,

cytoplasmic gene expression could prevent the risk of integration into the host genome [5,7,10,19–23]. This candidate reduces virulence and augments immune response, evidenced by elevated neutralizing antibody and IFN-γ production levels in murine and rabbit models. Genomic analysis has revealed significant deletions compared with those in the progenitor strain, indicating an enhancement in both the safety and efficacy profiles of KVAC103, making it a viable candidate for smallpox vaccination [15]. Concurrently, research has focused on using attenuated vaccinia virus vectors to develop vaccines against infectious diseases and therapeutic vaccines for cancer [24,25]. Recent advancements in vaccine development using KVAC103 as a viral vector have underscored its versatility and potential for addressing various infectious diseases. Notably, a bivalent vaccine targeting anthrax and smallpox was formulated by integrating the gene encoding the protective antigen of *Bacillus anthracis* into the KVAC103 genome [19]. This innovative approach utilizing KVAC103 demonstrates the adaptability of the platform for developing vaccines against different pathogens, including a promising TB vaccine candidate. KVAC103, an innovative third-generation smallpox vaccine candidate, was developed using the Lancy-Vaxina strain [15]. KVAC103 is a derivative of the vaccinia virus, closely related to the Lister and VACV107 strains, and demonstrates significant genomic alterations with 23 deleted and 4 truncated genes compared to VACV107. These alterations included the absence of virulence-related genes, such as the Bcl-2 homolog F1L, and deletions of the K1L and K3L genes, suggesting reduced virulence and cutaneous toxicity. Furthermore, KVAC103 shares 98.35% homology with MVA, indicating its potential use as a viral vector [15]. Research conducted in 2010 expanded the understanding of factors influencing MVA host range and virulence, revealing that the phenotypic traits of MVA are governed by not only known host range genes (K1L, C7L, C12L/SPI-1, E3L, and K3L), but also additional, yet unidentified, viral genes. Among the 31 open reading frames in the six large deletions of MVA, it is hypothesized that one or more uncharacterized genes significantly influence the virus's host range *in vitro* and reduce virulence in mammalian hosts, particularly in the parental chorioallantois vaccinia virus Ankara background [26]. Using advanced genetic engineering with a fully sequenced bacterial artificial chromosome clone of chorioallantois vaccinia virus Ankara, we sequentially introduced six major deletions into KVAC103 to avoid unwanted mutations. Interestingly, this only moderately reduced the overall virulence and significantly impaired replication in rabbit T-cells. The insights into the genetic modifications of KVAC103, particularly its relationship with MVA and its distinct genomic traits, make it a promising candidate for vaccine development. The reduced virulence and specific host range of KVAC103, coupled with its high homology to MVA, suggest its potential advantages in safety and efficacy as a viral vector. This study aims to explore the potential of KVAC103, enhanced through advanced genetic engineering techniques, as a versatile vaccine platform for TB and other infectious diseases. The antigen 85 complex consists of the genes *fbpA*, *fbpB*, and *fbpC* encoding Ag85A, Ag85B, and Ag85C, respectively, which display enzymatic activity in the mycobacterial envelope. The Ag85 complex is associated with human TB and comprises 20%–30% of the constitutive proteins present in the supernatant in short-term culture [7,20,23]. In particular, Ag85A (*fbpA*) and Ag85B (*fbpB*) have been selected as targets for developing TB vaccine candidates, and at least seven vaccines are currently in the clinical phase including Ag85A and Ag85B [7,10,15,19–28]. The cell wall of Mtb is a rigid structure composed of mycolic acid, D-arabino-D-galactan, and peptidoglycan [29]. Among these structures, the Ag85 complex plays a pivotal role in virulence, consisting of 85A, B, and C, which are highly homologous in sequence and structure [30]. Ag85 complex is the most common protein secreted from cultures, and although the major components of Ag85 are secreted, a small amount of antigen remains on the bacterial surface. For *M. tuberculosis*, the major components of Ag85 are Ag85A and 85B, which are about 60% of the total culture protein, with the Ag85B component being slightly more abundant compared to Ag85A [31].

The Ag85 complex is the primary virulence factor for *M. tuberculosis* pathogenesis, which interacts with fibronectin binding protein (Fbp) upon mycobacterial entry into the cell and binding to the mucosal surface. This interaction aids mycobacterial entry into the phagosomal components of the host cell and ultimately enables survival after intracellular infection [30]. Ag85B is a major extracellular cellular protein that is secreted within the first 3 days of infection and is therefore highly associated with early infection [32].

We generated a recombinant virus expressing the Ag85B protein based on the attenuated vaccinia virus KVAC103 (rKVAC85B) and evaluated its efficacy for two types of vaccination, the prime-boost of rKVAC85B and BCG prime-rKVAC85B boost. Additionally, mice were challenged via aerosol using two strains, *M. tuberculosis* H37Rv and the hyper-virulent strain HN878, to evaluate their protective effects on the bacterial load in the lungs.

## Materials and methods

### Animals and ethics statement

Four-to-five-week-old female C57BL/6 mice were procured from DooYeol Biotech (Seoul, Korea). Upon arrival, the mice were allowed to acclimatize for 1 week under controlled environment of the animal care facility at the Korea Disease Control and Prevention Agency. The subsequent experimental procedures strictly adhered to the ethical guidelines of the Institutional Animal Care and Use Committee of the Korea Disease Control and Prevention Agency under the approved protocol (permit number: KCDC-IACUC-02-042).

### Bacterial strains

*Mycobacterium bovis* BCG Pasteur, *M. tuberculosis* H37Rv, and *M. tuberculosis* HN878 strains were cultured in Middlebrook 7H9 broth (BD Biosciences, NJ, USA) enriched with 0.02% glycerol (Sigma, MO, USA), 0.05% Tween 80 (Sigma), and 10% albumin dextrose catalase (BD Biosciences) until they reached an optical density of 0.6–0.8. The cultures were centrifuged at 4290 × *g* for 20 min and washed thrice with PBS. To reduce clumping, the pellets were homogenized using a 26-gauge syringe. Colony-forming units (CFUs) for each strain were determined by plating the cells on Middlebrook 7H10 agar (BD Biosciences) supplemented with 0.05% glycerol and 10% oleic acid albumin dextrose catalase (BD Biosciences) and incubating at 37°C for 3–4 weeks.

### Generation and verification of recombinant KVAC103 expressing Ag85B

Ag85B, codon-optimized for human expression, was cloned into the vaccinia virus shuttle vector pVVT-1, which included the TK-R and TK-L regions, for homologous recombination to generate rKVAC85B (Fig 1A). Vero cells (ATCC CCL-81) were cultured in Dulbecco's modified Eagle medium (Gibco, NY, USA) supplemented with 10% fetal bovine serum (FBS; Gibco) and antibiotics. The medium was replaced with Opti-MEM (Gibco) containing 2% FBS to produce KVAC103 expressing GFP (rKVAC-GFP) and rKVAC85B cells. Cells adapted to 2% FBS were infected with rKVAC-GFP at a multiplicity of infection of 0.02 for 2 h in an environment maintained at 37°C and 5% $CO_2$, and then transfected with 1.5 µg pVVT-Ag85B using Lipofectamine 2000 (Thermo Fisher, CA, USA). After 5 h, the supernatant was replaced with Opti-MEM containing 2% FBS, and plaque isolation was performed after 2 days to select the recombinants, confirmed via polymerase chain reaction (PCR) and western blotting. For western blotting, polyclonal antibodies specific for *M. tuberculosis* Ag85B (1:1000; Abcam, Cambridge, UK) were used to detect the expressed proteins. The PCR confirmation of rKVAC85B was performed using specific primers: forward 5′-tttgaagcattggaagcaact-3′ and reverse 5′-acgttgaaatgtcccatcgagt-3′. The attenuated vaccinia viruses, KVAC103 and rKVAC85B, were grown in Opti-MEM containing 2% FBS without antibiotics for 3 days and titrated using the plaque-formation assay. To verify rKVAC85B, 1ul of plaque supernatant was used as a template and run for initial-denaturation, 95°C 5min, 30 cycles of 95°C 30sec-57°C 30sec-72°C 3min for amplification, and 72°C 5min for final extension.

### Animal immunization and *M. tuberculosis* H37Rv and HN878 infection via aerosol

To evaluate the efficacy and immunogenicity of rKVAC85B, both alone and in combination with BCG, mice were subjected to two distinct immunization strategies. The first strategy involved direct prime-boost inoculations with rKVAC85B administered at 3-weeks intervals to assess the response to rKVAC85B over a sustained period. The second strategy involved

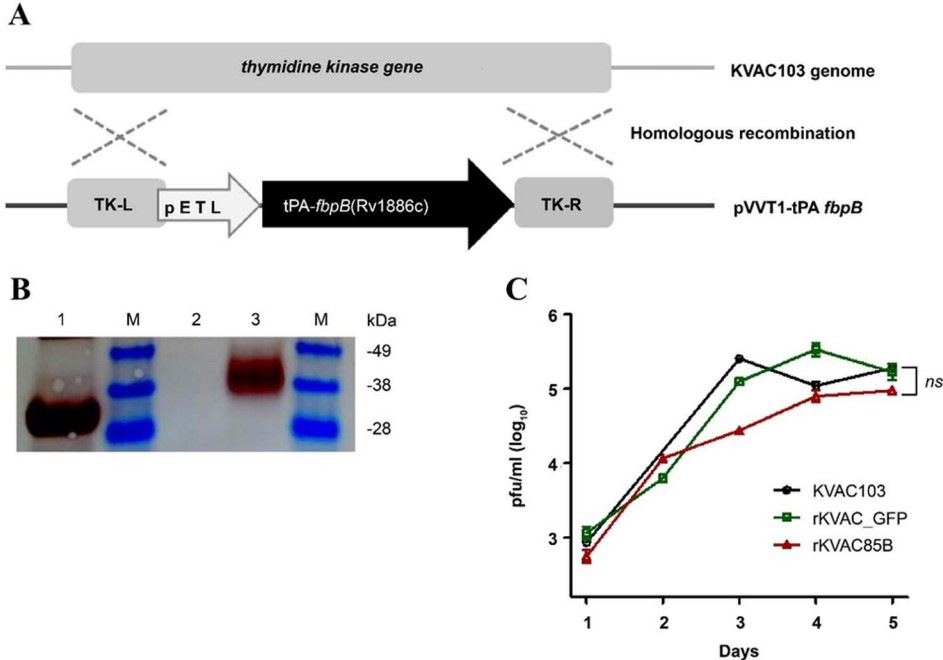

**Fig 1. Construction and expression of rKVAC85B in Vero cells. (A)** Schema illustrating the generation of recombinant KVAC virus expressing the *M. tuberculosis* Ag85B protein. To create rKVAC85B, a codon-optimized Rv1886c was inserted into the KVAC genome, containing TK-L and TK-R regions, through homologous recombination. TK: thymidine kinase. **(B)** Confirmation of Ag85B protein expression was conducted through western blot analysis using a rabbit polyclonal antibody against Ag85B (Abcam, Cambridge, UK). The purchased Ag85B protein (Abcam, Cambridge, UK) was used as the positive control, whereas the lysate obtained from the rKVAC-GFP infected group was used as the negative control. M: protein marker; Line 1: positive control (purified Ag85B protein), Line 2: negative control (Vero cell lysate), Line 3: rKVAC85B (Vero cell-infected lysate). **(C)** Comparative growth analysis was performed for the rKVAC85B and rKVAC-GFP groups. The growth titrations of these two viruses were assessed in Vero cells at a multiplicity of infection of 0.1, and the supernatants of the virus culture media were collected and harvested at 24 h intervals. Virus titrations were determined using a plaque assay.

BCG prime followed by an rKVAC85B boost, with rKVAC85B administered 10 weeks after the initial BCG inoculation at a dosage of $2 \times 10^5$ CFUs per mouse. Notably, in the BCG prime-rKVAC85B boost strategy, BCG was not administered as an additional boosting inoculation after the initial prime—a one-time BCG prime followed by an rKVAC85B boost without further BCG administration. Immunogenicity assessments were performed 1 week after the final immunization at 15 weeks of age. Five groups of mice were euthanized via $CO_2$ inhalation. The immune response was evaluated by analyzing lung lymphocytes and splenocytes using an ELISPOT assay, intracellular cytokine staining (ICS), IgG titer quantification, and *Mycobacterium* growth inhibition assay (MGIA).

To challenge with *M. tuberculosis*, a separate group of mice (n = 5) was exposed to the H37Rv and HN878 strains 3 weeks after the final immunization. Infection was performed using a Glas-Col aerosol generator (Glas-Col LLC., IN, USA) to ensure a consistent initial dose of 100–200 CFUs per mouse. To ascertain the initial bacterial load, the lung tissues from the infected mice were homogenized in DPBS the day after exposure. Homogenates were cultured on 7H10 agar plates for 3–4 weeks at 37°C to facilitate colony formation. Eight weeks post-exposure, the mice were euthanized, and the bacterial burdens in the lungs and spleen were determined by culturing on 7H10 agar for an additional 3–4 weeks. For histopathological analysis, tissue specimens were initially fixed in 10% neutral-buffered formalin solution. Subsequently, they were encapsulated in paraffin, sectioned to slices of thickness 4–5 μm, and stained using hematoxylin and eosin to enhance microscopic examination. For morphometric lesion and inflammation analyses, the lung sections were stained

with H&E and the appearance of the lesion was checked from low magnification to high magnification (20x~400x). The area where the alveolar wall was thickened due to inflammatory cell infiltration and proliferation of lung epithelial cells was determined, and the grade of the lesion due to tuberculosis infection was determined. The total area and the area of inflammatory cell infiltration thought to be a tuberculosis lesion were measured in the lung tissue photograph taken at low magnification.

## Measurement of antigen-specific IgG and bead-based analysis of cytokines

Enzyme-linked immunosorbent assay (ELISA) was conducted to quantify antibodies in a 1:100 ratio diluted sera, with a focus on IgG level assessed against plates coated with Ag85B recombinant protein antigen (100 ng/well; Abcam, Cambridge, UK), and the optical density at 450 nm was measured. To evaluate the immune response further, cytokines and chemokines were quantified in immune cells isolated from the lung and spleen tissues. The cells were plated at a density of $5 \times 10^5$ cells/well and stimulated with Ag85B peptides (79 peptides of Ag85B, 15mers with 11 amimo acid overlap, 100ng/well, JPT, Berlin, Germany) for 36 h. The subsequent bead-based ELISA, conducted in triplicate using the cell supernatant, enabled the measurement of a 10-cytokine panel: Th1/Th2/Th/17 related interferon-gamma (IFN-γ), interleukin-2 (IL-2), tumor necrosis factor-alpha (TNF-α), interleukin-17A (IL-17A) and, pro-inflammatory related interleukin p40 (IL-12p40), interleukin-12p70 (IL-12p70), interleukin-6 (IL-6), monocyte chemoattractant protein-1 (MCP-1), granulocyte-macrophage colony-stimulating factor (GM-CSF), and anti-inflammatory related interleukin-10 (IL-10). Comprehensive cytokine profiling was performed using a customized Bio-Plex mouse cytokine 10-plex assay (Bio-Rad, Hercules, CA, USA).

## INF-γ ELISPOT assay and intracellular cytokine staining

Isolated cells from the lung and spleen were stimulated using a purchased peptide mixture of Ag85B, which was subsequently subjected to ELISPOT (for IFN-γ-released cells) and intracellular staining (for T-cells secreting three types of cytokines). For the IFN-γ ELISPOT assay, $5 \times 10^5$ cells/well were seeded in plates coated with anti-mouse IFN-γ mAb. The assay was performed following the protocol provided by the manufacturer (BD Bioscience, San Jose, CA, USA). Each isolated cell was stimulated with an Ag85B peptide mixture (100 ng/well, JPT, Berlin, Germany) for 6 h, stained with antibodies developed against surface markers, and the results were analyzed using the FlowJo software (BD Biosciences, San Jose, CA, USA).

## MGIA

To generate a standard curve, a series of dilutions was prepared using *M. bovis* BCG 1173P2 ($1 \times 10^8$ CFU/mL) and phosphate-buffered saline with Tween 80. These dilutions, spanning seven 10-fold increments, were inoculated in MGIT tubes (BD Biosciences) containing MGIT PANTA (BD Biosciences MGIT growth supplement). Time to detection (TTD) was measured using an MGIT 960 instrument, which monitored the tubes for positivity at hourly intervals. Additionally, the dilutions were cultured on 7H10 plates to enumerate CFUs. To assess the TTD of isolated splenocytes, immunized splenocytes ($1 \times 10^6$ cells/300 μL per well) were first resuspended in RPMI1640 supplemented with HEPES and L-glutamine (HyClone, Waltham, MA). The cell suspension was then transferred to a 12-well plate, and *M. bovis* BCG (50 CFUs/300 μL) was added to the same medium. This co-culture was maintained for 4 days at 37°C in a 5% $CO_2$ incubator. Following the co-culture period, the splenocyte-BCG mixture was transferred to 1.5-mL tubes and centrifuged at 16,260 × *g* for 10 min at 4°C. The supernatant was carefully removed, and the pellet was resuspended in 600 μL tissue-culture-grade water. The resuspended pellet was inoculated into MGIT tubes containing MGIT PANTA. The tubes were placed in an MGIT 960 instrument and continuously monitored until they were turned positive.

## Statistical analysis

Statistical analyses to assess the significance of the results were conducted using the GraphPad Prism software. For multiple group comparisons, a one-way ANOVA followed by the Tukey–Kramer multiple comparison test was used. The significance levels were set as follows: $*p < 0.05$, $**p < 0.01$, $***p < 0.001$; "ns" denotes lack of statistical significance.

## Results

### Construction and growth analysis of KVAC103 expressing *M. tuberculosis* Ag85B (rKVAC85B)

Human codon-optimized Ag85B was successfully cloned into the shuttle vector pVVT1. This construct was then transfected into the attenuated vaccinia virus KVAC103 expressing GFP (rKVAC-GFP), producing recombinant virus rKVAC85B. rKVAC85B generation was verified through plaque assays on Vero cells and confirmed via PCR (data not shown). The expression of rKVAC85B was further validated via western blot analysis using monoclonal antibodies against the antigen 85B (Fig 1B). A comparative analysis of plaque-forming units between rKVAC-GFP and rKVAC85B indicated a minor delay in the growth of rKVAC85B at 2–3 days post-infection. However, by day 5, the growth rate of rKVAC85B was similar that of rKVAC-GFP (Fig 1C).

### Immune responses of rKVAC85B prime-boost

To explore the potential of rKVAC85B to replace the traditional BCG vaccine, we evaluated the immunogenic response elicited by a prime-boost immunization regimen in C57BL/6 mice. The mice were administered two consecutive injections of rKVAC85B at 3-weeks intervals. Subsequent ELISA employing the Ag85B recombinant protein as the coating antigen demonstrated a significant elevation in antigen-specific IgG titers in the sera of rKVAC85B-immunized mice. This elevation was markedly higher than that observed in the control groups that received either PBS or traditional BCG vaccination (Fig 2B). Further investigation of cellular immunity revealed an enhanced response in rKVAC85B-immunized mice. The IFN-γ ELISPOT assays conducted on splenic and pulmonary lymphocytes, upon stimulation with a comprehensive mixture of Ag85B peptides, showed an increase of over 2-fold and 2.5-fold in IFN-γ-secreting cells within the spleen and lungs, respectively, compared with those in the BCG-immunized group (Fig 2C). A more nuanced analysis of T-cell functionality was performed through intracellular cytokine staining after 36-h stimulation with the Ag85B epitope peptide mixture. The flow cytometry analysis identified a substantial fraction of CD4+ and CD8+ T lymphocytes capable of simultaneously secreting multiple cytokines, specifically, IL-2, IFN-γ, and TNF-α. These multifunctional T-cells comprised approximately 10% CD4+ and 11% CD8+ T-cells in the spleen, a notable increase compared with those in the BCG control group (Fig 2D and 2E).

### Antigen-specific IgG and IFN-γ-secreting cells induced via BCG prime-rKVAC85B boost

In response to the diminished efficacy of the BCG vaccine observed following infant immunization, in this study, we evaluated potential immunogenic enhancement through a BCG prime-boost strategy, utilizing rKVAC85B as the booster. We compared the immunological responses elicited by a BCG-prime followed by an rKVAC85B boost following the immunization regimen detailed in the Materials and Methods. The mice were euthanized 10–14 days after the final rKVAC85B inoculation, and their blood (to obtain serum), spleens, and lungs were collected for analysis. The ELISA quantitatively demonstrated a significant increase in antigen-specific IgG levels in the rKVAC85B group serum compared with that in both PBS control and BCG-only groups, as shown in Fig 3B. For an in-depth evaluation of cell-mediated immunity, we meticulously homogenized the splenic and pulmonary tissues to isolate the resident immune cells. These cells were then stimulated for 36 h with a carefully selected Ag85B epitope peptide mixture to assess IFN-γ secretion. The resulting analysis revealed a substantial increase in the proportion of IFN-γ-secreting lymphocytes in the rKVAC85B group, with an approximately 20-fold enhancement in the spleen and a 15-fold increase in the lung tissues relative to that in the BCG-only group (Fig 3C). These findings underscore the synergistic effect of the BCG-prime and rKVAC85B-boost in

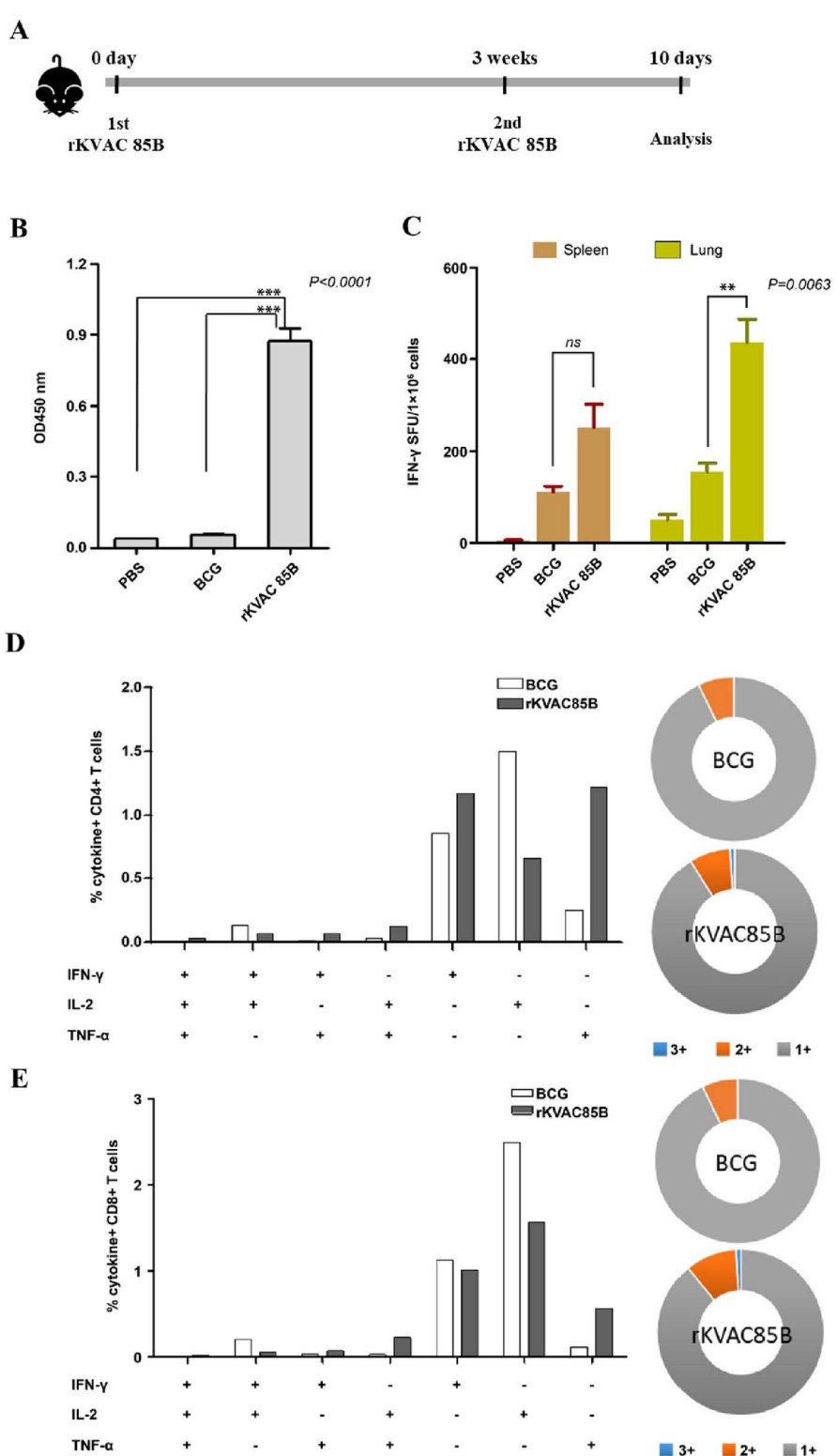

**Fig 2. Immune responses of rKVAC85B prime-boost immunized mice. (A)** Scheme of rKVAC85B prime-boost immunazation. rKVAC85B (5 × 10⁷ pfu/mouse) was subcutaneously inoculated 2 dose 3 weeks intervals and then immunized mice were sacrificed after 10-14 days from 2nd inoculation of rKVAC85B. **(B)** IgG antibody response to M. tuberculosis Ag85B antigen. Serum levels of antigen-specific IgG were quantified using ELISA, with

microtiter plates coated with recombinant Ag85B protein (Abcam, 100 ng/well) to assess humoral immunity post-vaccination. **(C)** Frequency of IFN-γ-secreting T-cells in the lungs and spleens as measured using ELISPOT. Lymphocytes isolated from the lungs and spleens of vaccinated mice were stimulated ex vivo with *M. tuberculosis* Ag85B peptide mixture (JPT, 100 ng/well). Spot-forming units per $1 \times 10^6$ cells were enumerated to determine the Ag-specific T-cell responses. **(D)** Polyfunctionality of CD4+ T-cells. The chart represents the percentage of CD4+ T-cells secreting different combinations of IFN-γ, TNF-α, and IL-2 upon stimulation with Ag85B protein. The pie charts illustrate the distribution of T-cells based on cytokine-secretion profiles: single (gray), double (orange), and triple (blue) cytokine producers. **(E)** Polyfunctionality of CD8+ T-cells. Similar to panel **(D)**, this chart depicts the percentage of CD8+ T-cells secreting cytokines IFN-γ, TNF-α, and IL-2, with the accompanying pie charts showing the proportions of mono-, bi-, and tri-cytokine-secreting cells.

**A**

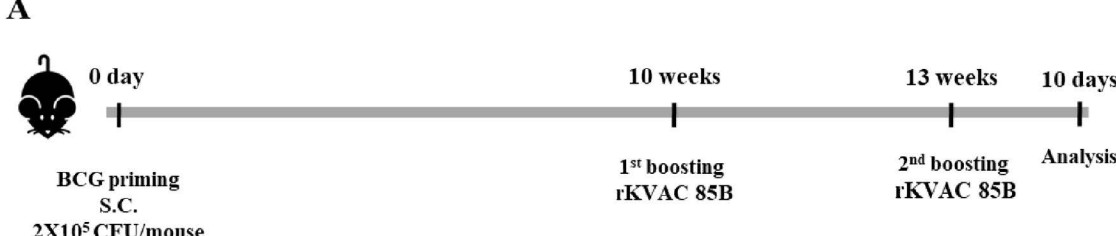

**B** **C**

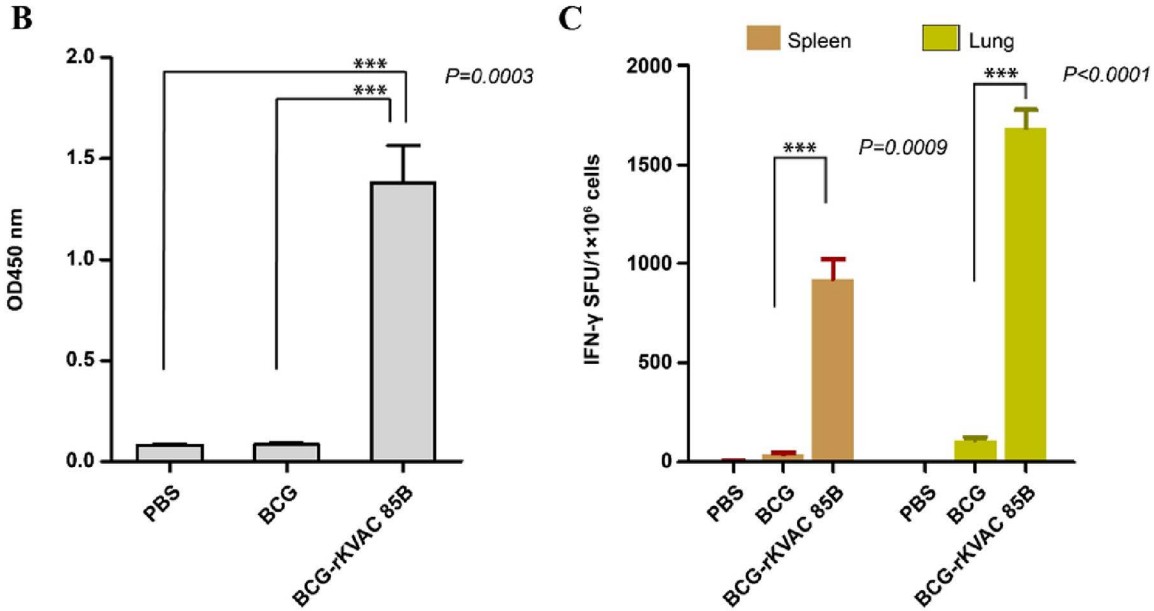

**Fig 3. IgG titers and IFN-γ ELISPOT responses were assessed via BCG prime-rKVAC85B boost immunization in mice. (A)** Scheme of BCG prime-rKVAC85B boost immunization. To induce BCG immune responses, mice were inoculated BCG $2 \times 10^5$ CFU/mouse and then maintained for 10 weeks. The mice were subcutaneously inoculated rKVAC85B ($5x\ 10^7$ pfu/mouse) 2 dose 3 weeks intervals and then immunized mice were sacrificed after 10-14 days from 2nd inoculation of rKVAC85B. **(B)** Determination of Ag85B-specific IgG titers. Sera from the immunized mice were diluted at 1:200 and applied to ELISA plates coated with the Ag85B antigen (Abcam, 100 ng/well) to measure specific IgG titers. The optical density was recorded at 450 nm (OD_450 nm). **(C)** IFN-γ ELISPOT assay to identify activated T-cells. The ELISPOT method was used to quantify IFN-γ-secreting T-cells in the spleen and lung tissues. Cells were stimulated with Ag85B peptides (JPT, 100 ng/well) for 36 h, and the frequency of IFN-γ-producing cells was measured. Notably, the lung tissue from the rKVAC85B-boosted mice showed a significant elevation in IFN-γ-secreting cells (***$p < 0.0001$) compared to the spleen, which displayed a less marked increase (***$p = 0.0009$).

significantly augmenting both humoral and cellular immune responses, particularly in producing antigen-specific IgG and activating T-cells.

## Polyfunctional T-cell activation via BCG prime-rKVAC85B boost

To assess the polyfunctional T-cell responses induced via BCG prime-rKVAC85B boost vaccination, immune cell populations isolated from lung and spleen samples were subjected to multiparametric flow cytometry. Cells were stained with a panel of antibodies specific for key cytokines: IFN-γ, IL-2, and TNF-α. This analysis aimed to identify a subset of T-cells capable of simultaneously secreting multiple cytokines. In the cohort receiving the BCG prime-rKVAC85B boost, 13% of the CD4+ T-cell population in the spleen exhibited a tri-cytokine secretion profile (IFN-γ+/IL-2+/TNF-α+). Furthermore, 19% of these cells demonstrated bi-cytokine positivity across various combinations (IFN-γ+/IL-2+, IFN-γ+/TNF-α+, and IL-2+/TNF-α+). In contrast, the group receiving only BCG vaccination exhibited a significantly lower frequency of polyfunctional CD4+ T-cells, with only 5% showing triple or double cytokine positivity, as shown in Fig 4Ai. Similarly, the cytokine secretion pattern of the CD8+ T-cells was mirrored in splenic CD4 + T-cells, which did not show significant polyfunctionality in either mouse group (Fig 4Aii).

In the lung tissue samples, the BCG prime-rKVAC85B boost group demonstrated enhanced T-cell polyfunctionality; 10% of CD4+ T-cells were tri-cytokine-positive and 20% displayed bi-cytokine positivity. In the BCG-only group, these figures were 7% and 3% for tri- and bi-cytokine-positive CD4 + T-cells, respectively (Fig 4Bi). Among the CD8+ T-cell populations in the lungs, the BCG prime-rKVAC85B boost group exhibited 9% tri-cytokine and 12% bi-cytokine positivity, compared to 3% and 11%, respectively, in the BCG-only group (Fig 4Bii). These findings delineate a significant enhancement in polyfunctional T-cell responses, particularly in CD4+ subsets, following BCG prime-rKVAC85B boost vaccination, highlighting its potential to elicit a more robust and diverse immune response.

## BCG prime-rKVAC85B boost enhances lung cytokine response

Cytokine secretion by T-cells in the lung and spleen tissues was quantified following a 36-h stimulation with an Ag85B peptide mixture using a bead-based ELISA approach. This analysis focused on a range of cytokines, including both pro- and anti-inflammatory markers (GM-CSF, IL-12p40, IL-12p70, IL-6, MCP-1, and IL-10) and cytokines associated with Th1 and Th17 responses (IL-17A, IL-2, TNF-α, and IFN-γ), to evaluate the antigen-specific cytokine profile induced by the BCG prime-rKVAC85B boost. The results showed that the levels of all tested cytokines were elevated more than two-fold in the lungs following BCG prime-rKVAC85B boost immunization compared with those in the BCG-only group. Notably, eight of these cytokines, including IL-17A, IL-2, TNF-α, and IFN-γ, exhibited a significant increase in their levels ($p <$ 0.001, $p <$ 0.01) in the lung (Fig 5). Similar results were observed when stimulation was conducted with purified protein derivative (PPD) (S1 Fig). This consistency further supports the robustness of the antigen-specific immune response elicited by the BCG prime-rKVAC85B boost, indicating a broad and effective activation of T-cells capable of responding to different tuberculosis antigens.

## Growth inhibition by BCG prime-rKVAC85B boost, as determined using MGIA

The TTD values for *M. bovis* BCG 1173P2 based on different CFUs were measured using the MGIT method. The coefficient of determination ($R^2$) for the standard curve was significant at 0.9638, confirming the robustness of the results, as depicted in Fig 6A. Notably, splenocytes harvested from mice immunized with BCG demonstrated a significant and superior reduction in CFUs compared with that in the PBS control group ($p = 0.0029$, Fig 6B). Interestingly, splenocytes obtained from mice subjected to BCG prime-rKVAC85B boost immunization exhibited a substantially greater reduction in CFUs than those from mice of both PBS control group ($p = 0.0006$, Fig 6B) and BCG-only immunization group ($p = 0.00349$, Fig 6B). This difference was approximately 0.5 CFU $\log_{10}$ reduction. Consequently, the results indicated a significant decrease in *M. bovis* BCG growth following the BCG prime-rKVAC85B boost immunization regimen.

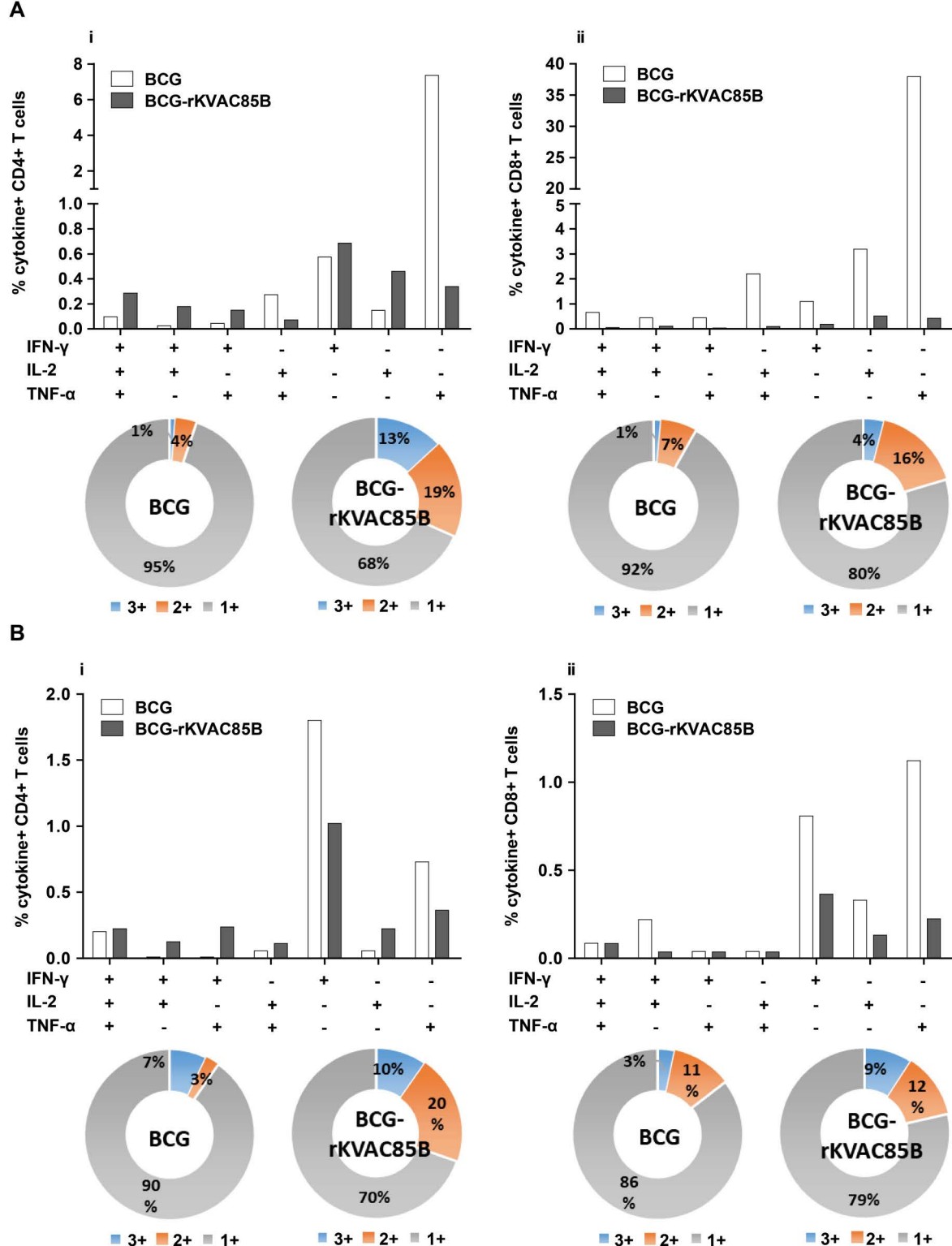

**Fig 4. Polyfunctional T-cell responses via BCG prime-rKVAC85B boost immunization in mice.** Polyfunctional T-cell profiling was conducted to assess the capacity of CD4+ and CD8+ T-cells to secrete the cytokines IFN-γ, TNF-α, and IL-2 following antigenic stimulation. **(A)** Frequency of polyfunctional CD4+ (i) and CD8+ T-cells (ii) in the spleen. Data are presented as the percentage of cells secreting any combination of the three cytokines,

with the pie charts reflecting the cell proportion secreting mono-, bi-, or tri-cytokines. **(B)** Frequency of polyfunctional CD4+ T-cells (i) and CD8+ T-cells (ii) in the lungs. The pie charts show the distribution of T-cells based on their cytokine-secretion profiles, categorized by their ability to secrete mono-, bi-, or tri-cytokines.

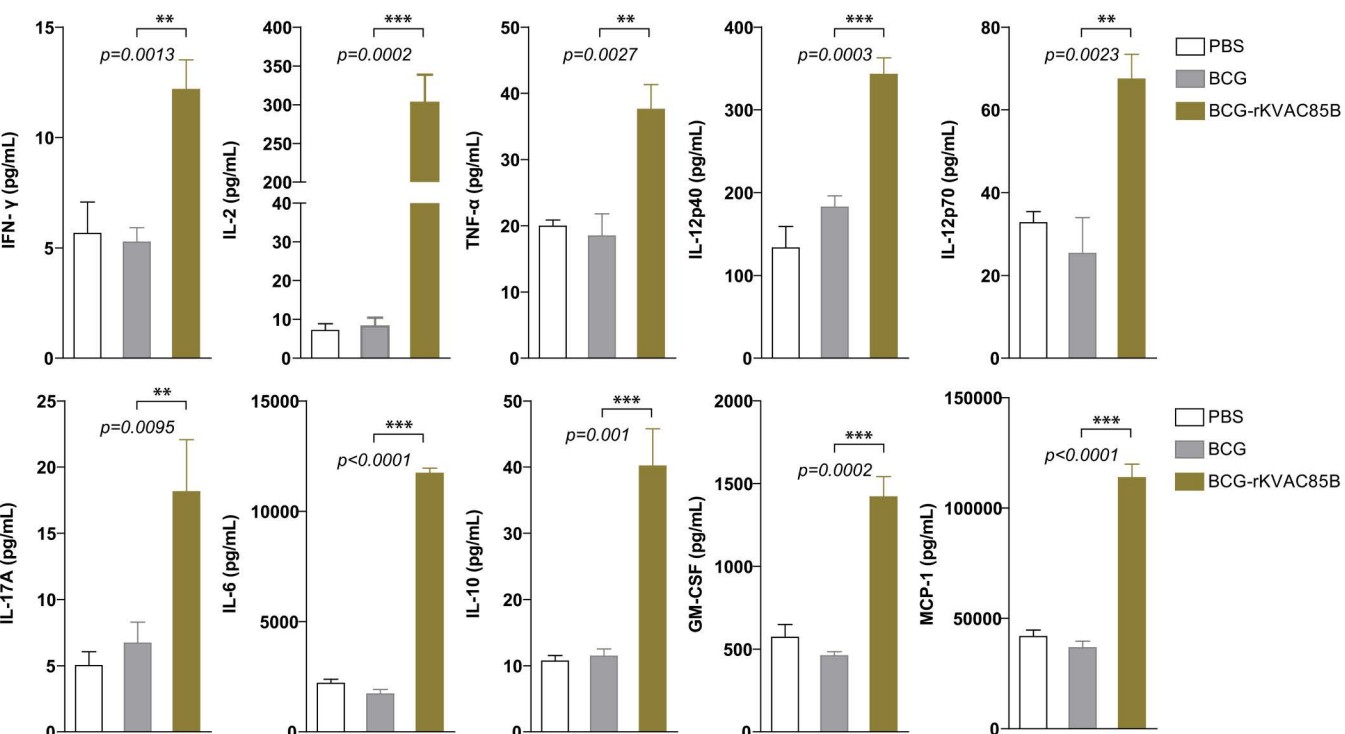

**Fig 5. Augmented pulmonary cytokine response following BCG prime-rKVAC85B boost.** Cytokine responses in the pulmonary compartment were quantified following a prime-boost vaccination regimen using bead-based ELISA. Lung cells were harvested and stimulated with antigen Ag85B peptides for 36 h to assess post-vaccination cytokine levels. This comparative analysis delineated cytokine induction across three groups: PBS control, BCG priming alone, and BCG priming followed by rKVAC85B booster. The cytokines measured include IFN-γ, IL-2, TNF-α, IL-12p40, IL-12p70, IL-17A, IL-6, IL-10, GM-CSF, and MCP-1. Results are expressed as mean ± standard deviation (SD), and levels of statistical significance are marked with asterisks: **$p < 0.01$, ***$p < 0.001$, indicating a significant increase in the cytokine levels in the BCG and rKVAC85B co-immunized groups.

## Efficacy of BCG prime-rKVAC85B boost against M. tuberculosis strains H37Rv and HN878

To evaluate the protective efficacy of rKVAC85B prime-boost and BCG prime-rKVAC85B boost regimens, we employed the laboratory standard strain *M. tuberculosis* H37Rv and the highly virulent HN878 strain for aerosol-mediated infection three weeks post-final vaccination. In the context of H37Rv challenge, the bacterial load in the lungs of mice vaccinated with BCG prime-rKVAC85B boost was reduced by 0.5 log ($p$ = 0.0088) compared to the unvaccinated control group (S1 Fig). Moreover, the response to the HN878 challenge demonstrated an even greater protective effect. Both lung and spleen bacterial counts dropped by more than 1.5 logs compared to the unvaccinated control group and by over 1 logs compared to the group that received only the BCG vaccine (Fig 7A and 7B). As depicted in Fig 7A and 7B, this significant reduction in bacterial burden against the HN878 strain underscores the effectiveness of the BCG prime-rKVAC85B boost regimen not only against the laboratory standard strain but also against highly virulent strains such as those of the W-Beijing lineage like HN878. Additionally, Fig 7C and 7D demonstrate results from H&E staining of lung samples for the detection of granulomas and inflammation, closely correlating with the CFU analysis findings.

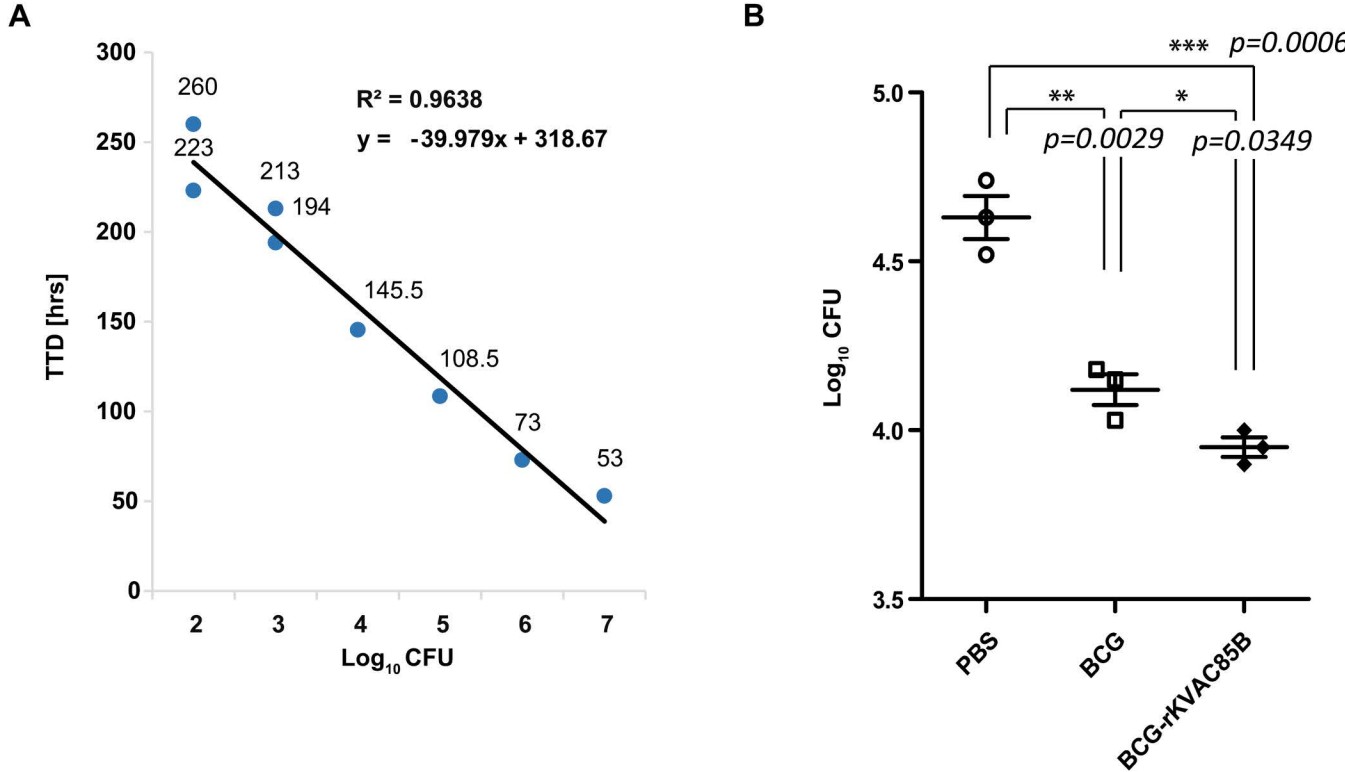

**Fig 6.** *In vitro* growth inhibition of *M. bovis* BCG with splenocytes isolated from BCG prime-rKVAC85B boost-immunized mice. **(A)** MGIA standard curve, depicting the relationship between the logarithm of colony-forming units ($\log_{10}$ CFUs) and time to detection (TTD) in days. The standard curve demonstrates a high coefficient of determination ($R^2$ = 0.9638), indicating a strong inverse correlation between Log_10 CFUs and TTD, as described by the linear regression equation y = −39.979x + 318.67. **(B)** One week after the final immunization, 1 × 10⁶ splenocytes from immunized mice groups were co-cultured with 50 CFUs of *M. bovis* BCG. Splenocytes were obtained from six control animals in each group, represented by an individual data point. The *p*-values of the differences were determined using a one-way ANOVA with Tukey's multiple comparison tests.

## Discussion

In this study, we developed rKVAC85B, a recombinant vaccinia virus expressing the Ag85B antigen, using the attenuated vaccinia virus KVAC103 as a novel vaccine platform [5,19]. Additionally, we evaluated the immune responses elicited by two vaccination strategies in a murine model: rKVAC85B prime-boost and BCG prime-rKVAC85B boost. Administration of the rKVAC85B vaccine in both vaccination models significantly increased the levels of antigen-specific IgG and IFN-γ, a cytokine essential for controlling infections caused by intracellular pathogens. The increase in the IFN-γ levels following rKVAC85B vaccination highlights the vaccine's potential to elicit a strong immune defense against intracellular infections. Quantitative analysis revealed that the antigen-specific IgG levels were approximately 1.58 times higher in the BCG-rKVAC85B group compared to the rKVAC85B alone group. Additionally, the IFN-γ ELISPOT responses indicated that the combined vaccination strategy elicited IFN-γ responses that were approximately 3.86-fold greater than those observed with the single rKVAC85B vaccination, as shown in Figs 2 and 3. Moreover, when comparing the BCG prime-rKVAC85B boost strategy to BCG alone, our data revealed a 30-fold increase in IFN-γ production, as evidenced in our experimental results (Fig 3C). This pronounced amplification of the IFN-γ response with the BCG prime-rKVAC85B boost underscores its enhanced efficacy in eliciting robust immune responses, crucial for effective immunological defense against tuberculosis. This comparison not only highlights the added benefit of the rKVAC85B boosting but also supports the necessity of enhancing BCG's immunogenic potential through supplementary strategies in vaccine development against intracellular pathogens.

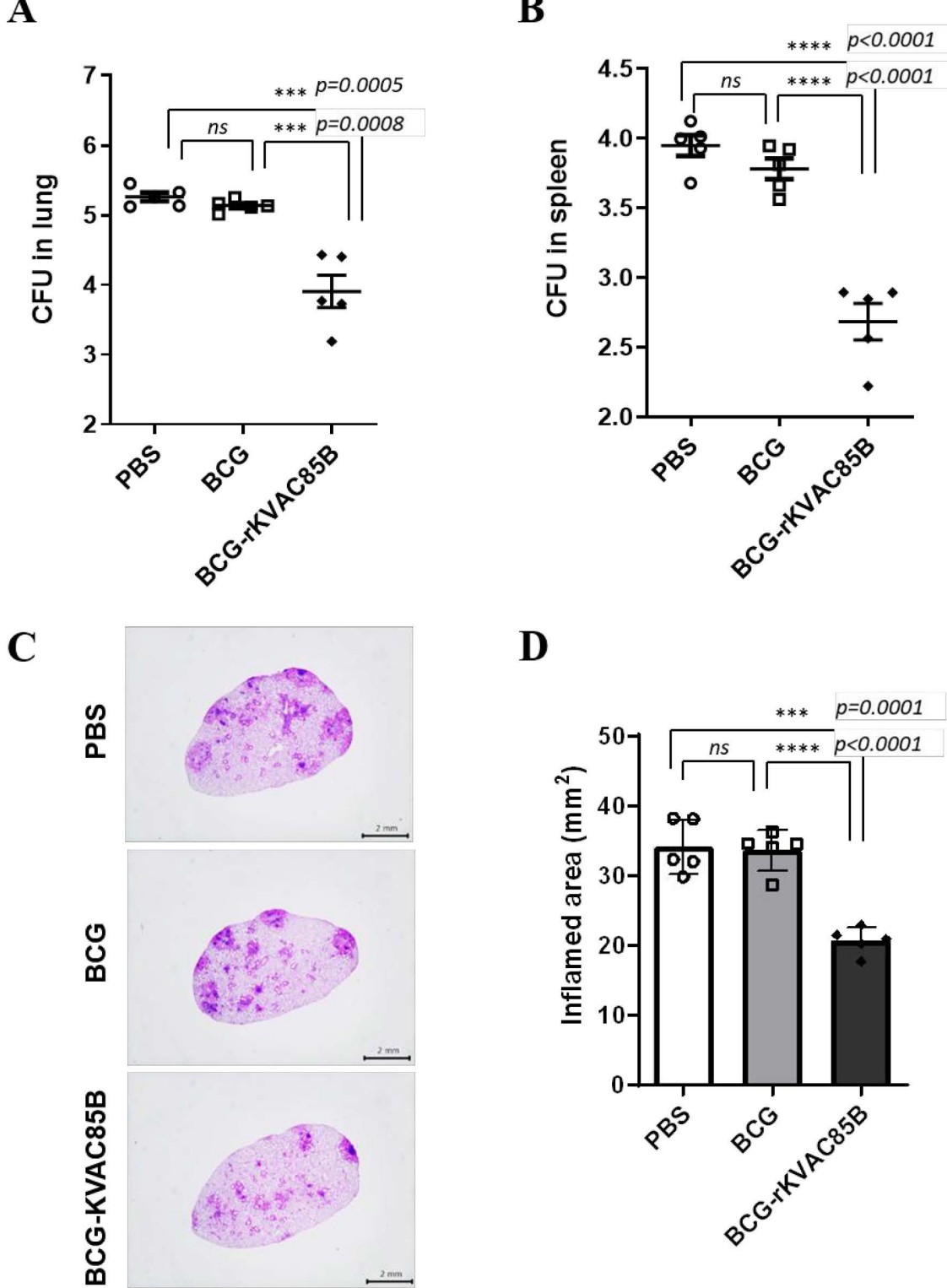

**Fig 7. Comparative efficacy of BCG prime-rKVAC85B boost immunization against *M. tuberculosis* HN878. (A)** Bacterial-load quantification in the lungs and **(B)** spleen of mice, following an immunization schedule of BCG priming and subsequent rKVAC85B boosts, was performed to determine the immunoprotective effects against two distinct strains of *M. tuberculosis* HN878 and H37Rv (S2 Fig). Mice were challenged with *M. tuberculosis* HN878 3

weeks after the last immunization dose. Eight weeks post-infection, CFUs in the lung tissue were enumerated to quantify the bacterial burden and infer the level of protection conferred by immunization. **(C)** and **(D)** display histological sections of the lung tissues stained with hematoxylin and eosin (H&E) post-infection with HN878 strains and compared to inflamed area, respectively. The scale bar represents 2 μm. Statistical significance of differences in CFUs between the groups was determined using the one-way ANOVA, with $p$-values indicating the level of significance, where *$p ≤ 0.05$, **$p ≤ 0.01$, and ***$p ≤ 0.001$. "*ns*" denotes not significant.

In the BCG prime-rKVAC85B boost group, there was a notable increase in the production of polyfunctional T-cells. These T-cells secreted cytokines critical for Th1 immunity, specifically IL-2, IFN-γ, and TNF-α. In both vaccination models, a marked increase in the levels of IFN-γ-secreting T-cells was observed. Additionally, the BCG prime-rKVAC85B boost strategy was particularly effective in inducing potent CD4+ and CD8+ multifunctional T-cell responses in the lungs, capable of secreting two or three cytokines (Fig 5). This finding is consistent with those of previous studies, highlighting the efficacy of viral vector-assisted immunization following BCG priming in enhancing cellular immunity [7,10,20,21]. Furthermore, our analysis revealed a substantial enhancement in the levels of cytokines related to Th1 and Th17 responses (IFN-γ, TNF-α, and IL-17) in the lungs (Fig 5). Although IL-2 is generally associated with Th1 and Th2 immune responses, the presence of IL-2 alone in our study did not provide sufficient evidence of a significant Th2 response. Instead, the observed cytokine profile and T-cell responses suggested that the rKVAC85B vaccine, especially in conjunction with BCG priming, primarily drives Th1 and Th17 immune responses. This comprehensive activation of the immune system underscores the potential of rKVAC85B as an effective component of TB vaccines, leveraging its ability to induce a broad and potent immune response.

Furthermore, our analysis has shown that following the administration of the rKVAC85B vaccine, there is a notable increase in IL-10 levels (Fig 5). This elevation in IL-10 is indicative of the vaccine's ability to orchestrate a balanced immune response, which is critical for moderating inflammation and preventing excessive tissue damage while maintaining an effective and controlled immune system. This regulatory response is characteristic of the immune modulation frequently observed with vaccinia virus vectors, such as rKVAC85B, which can provoke both pro-inflammatory and anti-inflammatory cytokine production, including IL-10 [33,34].

The ability to balance activation—ensuring it is sufficient to combat pathogens without inducing excessive inflammation—is crucial for the long-term effectiveness and safety of the vaccine. Such balanced immune responses prevent potential immunopathology and enhance the vaccine's safety profile, minimizing the likelihood of adverse effects in clinical settings [35,36]. The increased IL-10 following rKVAC85B administration therefore not only underscores the potential of rKVAC85B as an effective component of TB vaccines but also highlights the importance of further investigating the mechanisms by which this vaccine modulates immune responses.

Recent studies have underscored the pivotal role of IL-17 in the immune response to *M. tuberculosis*. This cytokine is particularly crucial in certain models of intracellular infection, where it aids in inducing the Th1 response by promoting IL-12 production. In TB, IL-17 mediates the protective effects of vaccines against *M. tuberculosis*. These findings suggest that the IL-17 pathway plays a major role in the primary immune response following *M. tuberculosis* infection [37]. Choi *et al*. demonstrated the effectiveness of the HSP90-E6/CIA05 vaccine in enhancing BCG-induced immunity against a hypervirulent *M. tuberculosis* HN878 strain in mice. This effect is primarily attributed to the expansion of IFN-γ/IL-17-producing lung cells. While IFN-γ-producing T-cells are necessary for TB defense, they are insufficient on their own. Instead, cells producing IFN-γ and IL-17 are crucial for effective protection [38]. These findings shed light on the essential role of IL-17 in TB immunity and offer new avenues for improving BCG-boosted vaccines.

In the current study, we assessed the *in vitro* growth-inhibition ability of BCG and a BCG prime-rKVAC85B boost using MGIA. Immunized splenocytes from the BCG prime-rKVAC85B boost group demonstrated significantly reduced CFUs compared with those from the BCG-only group (Fig 6B). This observation is consistent with the protective efficacy observed in the BCG prime-rKVAC85B boost group.

*Mycobacterium* growth inhibition, regarded as a surrogate marker for protection, has been used to establish in vitro functional assays that correlate with protection. Building on this premise, the research highlights the significant inhibition of BCG growth and the induction of a phenotypic shift in monocytes in mice by the RUTI vaccine, which is currently in clinical trial stages [39]. Additionally, studies have consistently shown that growth inhibition assays yield comparable results when using both BCG and *M. tuberculosis* as target bacteria for vaccines and therapeutics, supporting the notion that MGIA outcomes using BCG can meaningfully correlate with those using *M. tuberculosis* [40–42]. Furthermore, growth inhibition after co-culturing blood from non-infected individuals, active TB patients, and individuals with latent TB infection with both BCG and H37Rv was analyzed, observing similar influences on the growth of the two strains regardless of disease status. These findings further support the validity of using BCG in MGIA as an effective method for evaluating the efficacy of tuberculosis vaccines [43]. Furthermore, several reports have highlighted the advantages of this analytical method in the context of vaccine testing, allowing the longitudinal monitoring of immunological development in the same animal subjects [40,42].

To assess the protective effect of rKVAC85B against bacterial infection, immunized mice were exposed to both H37Rv and hypervirulent HN878 strains, and the bacterial load in the lungs was subsequently measured. In our study, immunization with rKVAC85B alone resulted in some reduction in the bacterial burden in H37Rv infection via aerosols; this decrease was not significantly greater than that in the BCG-immunized group. However, notably, the BCG prime-rKVAC85B boost markedly inhibited the growth of the hypervirulent HN878 strain (Fig 7) more effectively than that of the laboratory strain H37Rv (S2 Fig). This finding suggests that rKVAC85B can potentially enhance the vaccine efficacy of BCG, particularly against the more virulent *M. tuberculosis* strains.

The W-Beijing family of *M. tuberculosis*, the predominant genetic lineage of *M. tuberculosis* strains, is widespread globally, especially in East Asia [44]. Epidemiological studies have revealed that W-Beijing strains are isolated more frequently from BCG-vaccinated patients with TB than from non-vaccinated patients, suggesting a selective advantage of this genotype in the context of BCG vaccination [45]. Additionally, the BCG vaccination in mice offers less protection against isolates from the W-Beijing family than the standard laboratory *M. tuberculosis* H37Rv strain [44,46]. Notably, the clinical isolate HN878, a hypervirulent strain associated with increased mortality rates, exemplifies the challenges posed by virulent strains [47,48]. These observations emphasize the critical need for vaccine development strategies against the most prevalent and virulent *M. tuberculosis* genotypes, such as the W-Beijing family.

According to the Global TB Report 2021, TB in individuals older than 15 years accounted for 89% of the total cases reported in 2020 [1], which poses the demand for a novel TB vaccine acting as a BCG booster in adults. In the present study, we used KVAC103, a new attenuated vaccinia virus vector, to develop a novel TB vaccine candidate. Using KVAC103 to express Ag85B represents a promising development in TB vaccine research, particularly in adult populations. rKVAC85B, formulated using this innovative vector, has shown encouraging results by eliciting enhanced immune responses and notable protective effects in a BCG prime-boost regimen. Although these initial findings are promising, they suggest the potential of KVAC103 as an effective BCG booster and a possible frontrunner in the search for advanced TB vaccine solutions. This approach underscores the importance of exploring new avenues of virology for TB vaccine development, with KVAC103 showing considerable promise.

## Conclusions

In this study, we developed a recombinant vaccinia virus based on the attenuated KVAC103 strain, engineered it to express the Ag85B antigen, and assessed its immunogenicity using two vaccination models, rKVAC85B prime-boost and BCG prime-rKVAC85B boost. Compared to BCG alone, both vaccination strategies significantly enhanced the levels of antigen-specific IgG- and IFN-γ-secreting cells, and various cytokines. In addition, the *in vitro* analysis of *Mycobacterium* growth inhibition demonstrated a consistent immune-mediated pattern. We also evaluated the protective efficacy of the BCG prime-rKVAC85B boost vaccination against aerosol infections using two *M. tuberculosis* strains: H37Rv and

hypervirulent HN878. Our results revealed that both strains had a lower bacterial burden in the rKVAC85B group than in the BCG-only immunization group. Notably, the reduced bacterial load, particularly against the hypervirulent HN878 strain, suggests that rKVAC85B is a promising candidate for advancing TB vaccination strategies.

## Supporting information

**S1 Fig. Measurement of pulmonary cytokine response by PPD following BCG prime-rKVAC85B boost.** Cytokine responses in the pulmonary compartment were quantified following a prime-boost vaccination regimen using bead-based ELISA. Lung cells were harvested and stimulated with PPD (WHO international standard purified protein derivative of *Mycobacterium tuberculosis*, National Institute of Biological Standards and Control (NIBSC), 100 ng/well) for 36 h to assess post-vaccination cytokine levels. This comparative analysis delineated cytokine induction across three groups: PBS control, BCG priming alone, and BCG priming followed by rKVAC85B booster. The cytokines measured include IFN-γ, IL-2, TNF-α, IL-12p40, IL-12p70, IL-17A, IL-6, IL-10, GM-CSF, and MCP-1. Results are expressed as mean ± standard deviation (SD), and levels of statistical significance are marked with asterisks: $**p < 0.01$, $***p < 0.001$, indicating a significant increase in the cytokine levels in the BCG and rKVAC85B co-immunized groups.
(TIF)

**S2 Fig. Immunoprotective efficacy of BCG prime-rKVAC85B boosted against H37Rv.** Bacterial-load quantification in the lungs of mice, following an immunization schedule of BCG priming and subsequent rKVAC85B boosts, was performed to determine the immunoprotective effects against two distinct strains of *M. tuberculosis*. (A) H37Rv strain challenge: In an experimental setup similar to that in Fig 7 (A) HN878 strain challenge: CFUs in the lung tissues were counted 8 eight weeks to determine the strain-specific protective efficacy of the vaccine regimen. (B) display histological sections of the lung tissues stained with hematoxylin and eosin (H&E) post-infection with H37Rv strain. The scale bar represents 2 μm. Statistical significance of differences in CFUs between the groups was determined using the one-way ANOVA, with *p*-values indicating the level of significance, where $*p ≤ 0.05$, $**p ≤ 0.01$, and $***p ≤ 0.001$. "*ns*" denotes not significant.
(TIF)

**S1 Raw Data Gel Image. Original uncropped underlying bolt result (Corresponding to Fig 1).**
(TIF)

**S1 Table. Raw data file.**
(XLSX)

## Acknowledgments

We extend our gratitude to Professor Hyejon Lee from Yonsei University College of Medicine for their valuable participation in the MGIA.

## Author contributions

**Conceptualization:** Jong-Seok Kim, Hye-Sook Jeong.

**Data curation:** Sung-Jae Shin.

**Investigation:** Eunkyung Shin, Jin-Seung Yun, Young-Ran Lee, Hye-Ran Cha, Soo-Min Kim.

**Supervision:** Sang-Won Lee, Gyung Tae Chung, Dokeun Kim, Jung Sik Yoo, Jong-Seok Kim, Hye-Sook Jeong.

**Writing – original draft:** Eunkyung Shin.

**Writing – review & editing:** Jong-Seok Kim, Hye-Sook Jeong.

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
