## [Decision Letter · Decision Letter 0]

2 Jul 2024

PONE-D-24-18487Efficacy and immunogenicity of rKVAC85B in a BCG prime-boost regimen against H37Rv and HN878 Mycobacterium tuberculosis strainsPLOS ONE

Dear Dr. Jeong,

Thank you for submitting your manuscript to PLOS ONE. After careful consideration, we feel that it has merit but does not fully meet PLOS ONE’s publication criteria as it currently stands. Therefore, we invite you to submit a revised version of the manuscript that addresses the points raised during the review process.

**Please revise your manuscript as per suggestion of both the reviewers.  The comments should be addressed point wise indicating how the comment was addressed ( with section, page and line number). **

We look forward to receiving your revised manuscript.

Kind regards,

Syed Faisal, PhD

Academic Editor

PLOS ONE

2. To comply with PLOS ONE submissions requirements, in your Methods section, please provide additional information regarding the experiments involving animals and ensure you have included details on methods of anesthesia and/or analgesia, and efforts to alleviate suffering.

4. "We note that the grant information you provided in the ‘Funding Information’ and ‘Financial Disclosure’ sections do not match. 

5. We note that your Data Availability Statement is currently as follows: [All relevant data are within the manuscript.]

Reviewers' comments:

Reviewer's Responses to Questions

**Comments to the Author**

1. Is the manuscript technically sound, and do the data support the conclusions?

Reviewer #1: Partly

Reviewer #2: Partly

2. Has the statistical analysis been performed appropriately and rigorously? 

Reviewer #1: Yes

Reviewer #2: Yes

3. Have the authors made all data underlying the findings in their manuscript fully available?

Reviewer #1: Yes

Reviewer #2: Yes

4. Is the manuscript presented in an intelligible fashion and written in standard English?

Reviewer #1: Yes

Reviewer #2: Yes

5. Review Comments to the Author

Reviewer #1: The authors describes a novel vaccinal construct for vaccination against Tuberculosis. It is a novel attenuated vaccinia virus (KVAC103) expressing Ag85B (rKVAC85B). They analyzed the immunogenicity and the efficacy of the vaccine, also in a prime-boost inoculation strategies with BCG in a murine model of TB. the work is quite interesting even if not very original, unfortunately it has serious experimental limitations.

Major points:

1- In all the figures 2, 3, 4 e 5 only data "specific" for Ag85B are reported. The baseline responses (e.g. only culture medium) are missing. These controls are required to evaluate the baseline activation of cells and the increse/decrease due to Ag85B restimulation.

2- In Figure 2, the data related to the vaccination with BCG+rKVAC85B alone are missing. Is the prime with BCG enhances, reduces or leaves inaltered the Ag85B responses induced by rKVAC85B alone?

3- In all the figures 3, 4 e 5 the data related to the control vaccination with rKVAC85B alone are missing. An experiment in which all the various vaccine combinations are present at the same time is necessary to be able to make adequate comparisons. The data are required to explain the data showed in Figure 6 and Figure 7 (MGIA and in vivo MTB infection). Is the prime with BCG enhances, reduces or leaves inaltered the Ag85B responses induced by rKVAC85B alone?

4- Spleen cells of BCG-vaccinated mice are not good responder to Ag85B protein; ex vivo Ag85B restimulation can even inhibit the basal cytokine production by cells of BCG-vaccinated mice. Therefore, knowing not only the basal level of cytokine production but also that induced by stimulation with other MTB antigens, such as PPD, is essential to understand how the vaccine rKVAC85B modulates the entire BCG-mediated immune response.

5- Evaluation of Ag85B-specific immune responses after 1 week from the last immunization is not idoneous to reveal memory immunity, which is a priority to demonstrate the vaccine efficacy. Longer times need to be studied.6- In MGIA experiments (Figure 6), as well as in in vivo experiments (Figure 7) the data related to vaccination with rKVAC85B alone are missing. Is vaccination with rKVAC85B alone protective?

7- In MGIA experiments (Figure 6), the “potential” presence of BCG in spleen cells of BCG- or BCG+rKVAC85B-vaccinated mice, prior to in vitro BCG infection, needs to be evaluated. Do the spleens of mice vaccinated with BCG or BCG+rKVAC85B harbor live BCG at the time of organ explantation?

8- In Figure 7, why the CFU are higher in the lungs of mice infected with H37Rv instead of those infected with the hypervirulent HN878 strain? Furthermore, histology data should be graphed and an appropriate statistical analysis performed to demonstrate that the inflammation and area occupied by granulomas are higher in BCG- than in BCG+rKVAC85B-vaccinated mice.

9-In the Discussion, lines 398-417 and 420-444 are appropriate for the Introduction section.

10-in the Discussion, it is reported that “the increase in the IFN-γ levels following rKVAC85B vaccination highlights the vaccine's potential to elicit a strong immune defense against intracellular infections”, lines 449-450. Although the protective role of IFN-γ in TB is undoubted, many papers indicate that this cytokines needs to be balanced and its enhancement does not always correlate with better protection against Mtb infection. It is also true for the IFN-γ generated in response to Ag85B protein. This point should be discussed with greater attention also considering that there is no data in the work which supports the relationship between the increase in IFN-γ and protection.

Minor points

1- Add refs in the introduction section line 82

Reviewer #2: Reviewer's Comments on Manuscript PONE-D-24-18487

Title: Efficacy and Immunogenicity of rKVAC85B in a BCG Prime-Boost Regimen Against H37Rv and HN878 Mycobacterium Tuberculosis Strains

General Assessment:

The manuscript presents a study on the efficacy and immunogenicity of a recombinant vaccinia virus expressing Ag85B of M. tuberculosis (rKVAC85B) employing two prime-boost regimens against two strains of M. tuberculosis: standard laboratory H37Rv and the hypervirulent HN878 strains. The research is timely and addresses a critical need for improved tuberculosis vaccines, especially considering the variable efficacy of BCG in adults.

Strengths:

The study introduces a novel vaccine candidate using KVAC103, an attenuated vaccinia virus as vaccine vector to express Ag85B, which is of significant interest in the field of TB vaccine research. The focus on both H37Rv and hypervirulent HN878 strains enhances the study's relevance. The study adheres to ethical standards, with appropriate approvals and descriptions of animal welfare considerations.

Areas for Improvement:

1. Introduction:

While the existing introduction provides a wide background but could benefit from more detailed but specific information on the limitations of BCG and other current TB vaccines in clinical trial, and the specific advantages of using vaccinia virus vectors and the specific antigen (Ag85B) and its specific role in M. tuberculosis.

2. Materials and Methods:

The M&M section requires further comprehensive details. For instance:

(i) the description of the generation of rKVAC85B could include more details on the verification steps (e.g., PCR conditions, primer sequences, and agarose gel images) as supporting data, such that it should be easy to follow the protocols by other researcher or reader.

(ii) the authors should provide clear pictorial (cartoon) outlines of the immunogenicity and protective efficacy study protocols as part of the respective result figures, for greater clarity to the reader.

(iii) the methodology for the bead-based cytokine analysis could be expanded to include the rationale for selecting these cytokines.

(iv) while MGIA is an interesting and powerful ex vivo assay to measure the efficacy of cellular immunity in controlling bacterial growth, inclusion of BCG as the target bacteria reduces its significance in this study that targets to evaluate the efficacy of new vaccine against two virulent strains of M. tuberculosis. This requires appropriate rationale or redoing it with virulent M. tuberculosis.

3. Results:

The description of results and the respective figures need appropriate revision. For instance:

(i) the immunogenicity experiment seems to be conducted for all five groups together (Unvaccinated, BCG, rKVAC85B, rKVAC85B prime boost, and BCG- KVAC85B prime boost) as described in the M&M section (Line 150). However, results were depicted separately which makes it difficult to compare among all the groups. Animal numbers in each group for immunogenicity study was not mentioned, while for efficacy study it is mentioned as (n=5). Figure 2 C and D, and Figure 4 A and B bar diagrams should be provided with the error bars and appropriate statistics to show significance of the observed differences. The conc. of Ag85B protein that was used as the coating/stimulating agent not mentioned. Details of Ag85B epitope/peptides used for ELISPOT assay should be provided. Figure 5, IL-10 levels were depicted, typically this is a Th2 cytokine and known to enhance pathogen favoured response in TB, however, its heightened induction in the case of BCG prime rKVAC85B was not explained well. Information on number of samples, technical replicates should be provided.

(ii) as pointed out above, use of BCG as the target pathogen in the MGIA requires justification.

(iii) typically, in the mouse model of TB vaccine efficacy studies, both lungs and spleen CFU counts are provided. In this study, only lung CFU was shown. Speen CFU data highlights how the vaccination would prevent dissemination of infecting pathogen to distant lymphoid organ. It would be better to show both lung and spleen CFU data. Further, in the absence of details of lungs homogenisation and plating protocol (which/how many lobes for what purpose, weight of lungs) it is difficult to appreciate the CFU data presentation as Log10 CFU/ml as weight/size of lungs varies between vaccinated and unvaccinated animals. Figure 7 needs major revision, it is appropriate to depict CFU data as per lungs or per gram of tissue, CFU data for all the groups in the study is warranted, labelling of the unvaccinated and infected group as normal is technically incorrect, should be revised.

4. Discussion:

The discussion could be strengthened by more critically comparing the results with existing studies on TB vaccines, particularly those using similar viral vectors, and explaining why the vaccine under study could be superior. Discuss the conundrum of heightened L-10 response in the study. Also, highlight future research directions.

Minor Comments:

(i) Ensure consistency in the use of abbreviations throughout the manuscript.

(ii) A few typographical and grammatical errors need correction for clarity and readability.

(iii) Ensure all references are up-to-date and formatted according to the journal's guidelines.

6. PLOS authors have the option to publish the peer review history of their article (what does this mean? ). If published, this will include your full peer review and any attached files.

**Do you want your identity to be public for this peer review?** For information about this choice, including consent withdrawal, please see our Privacy Policy .

Reviewer #1: No

Reviewer #2: No

---

## [Author Response · Author response to Decision Letter 1]

18 Nov 2024

All specific responses to the reviewers` comments have been addressed in detail in the attached document.

Please refer to it for comprehensive explanations and revisions. Thank you.

---

## [Decision Letter · Decision Letter 1]

7 Jan 2025

PONE-D-24-18487R1Efficacy and immunogenicity of rKVAC85B in a BCG prime-boost regimen against H37Rv and HN878 Mycobacterium tuberculosis strainsPLOS ONE

Dear Dr. Jeong,

Thank you for submitting your manuscript to PLOS ONE. After careful consideration, we feel that it has merit but does not fully meet PLOS ONE’s publication criteria as it currently stands. Therefore, we invite you to submit a revised version of the manuscript that addresses the points raised during the review process.

We look forward to receiving your revised manuscript.

Kind regards,

Syed Faisal, PhD

Academic Editor

PLOS ONE

Journal Requirements:

Reviewers' comments:

Reviewer's Responses to Questions

**Comments to the Author**

1. If the authors have adequately addressed your comments raised in a previous round of review and you feel that this manuscript is now acceptable for publication, you may indicate that here to bypass the “Comments to the Author” section, enter your conflict of interest statement in the “Confidential to Editor” section, and submit your "Accept" recommendation.

Reviewer #1: (No Response)

Reviewer #2: All comments have been addressed

2. Is the manuscript technically sound, and do the data support the conclusions?

Reviewer #1: Partly

Reviewer #2: Yes

3. Has the statistical analysis been performed appropriately and rigorously? 

Reviewer #1: (No Response)

Reviewer #2: Yes

4. Have the authors made all data underlying the findings in their manuscript fully available?

Reviewer #1: Yes

Reviewer #2: Yes

5. Is the manuscript presented in an intelligible fashion and written in standard English?

Reviewer #1: Yes

Reviewer #2: Yes

6. Review Comments to the Author

Reviewer #1: Although some parts have been improved, the revised version of the manuscript did not address the various criticisms raised by the reviewers, especially those that would have required novel experiments. In particular, the authors did not provide comparable data across all the different groups investigated (PBS, rKVAC85B, rKVAC85B boosting, BCG and BCG-rKVAC85B) that would have allowed a complete and correct comparison to assert that boosting with rKVAC85B enhanced the vaccine efficacy of BCG. Also the lack of data on the basal levels of the various immunological parameters analyzed as well as the disparity of the vaccination protocols used do not allow to clarify what was actually modulated. Furthermore, the histology figures of the lungs of the infected mice do not seem very consistent with what has been reported in panel D of Figure 7.

Reviewer #2: The revised manuscript is substancially improved. Except some typographycal errors that requires rectification, there is no more comments.

7. PLOS authors have the option to publish the peer review history of their article (what does this mean? ). If published, this will include your full peer review and any attached files.

**Do you want your identity to be public for this peer review?** For information about this choice, including consent withdrawal, please see our Privacy Policy .

Reviewer #1: No

Reviewer #2: No

---

## [Author Response · Author response to Decision Letter 2]

21 Feb 2025

Reviewer-1`s Comments to Author

1. Although some parts have been improved, the revised version of the manuscript did not address the various criticisms raised by the reviewers, especially those that would have required novel experiments. In particular, the authors did not provide comparable data across all the different groups investigated (PBS, rKVAC85B, rKVAC85B boosting, BCG and BCG-rKVAC85B) that would have allowed a complete and correct comparison to assert that boosting with rKVAC85B enhanced the vaccine efficacy of BCG.

Reply: Thank you for your detailed comments and suggestions. We recognize the importance of providing comparative data across different groups to conclusively demonstrate that rKVAC85B boosting enhances the efficacy of the BCG vaccine, as you pointed out.

We conducted a comparative analysis of the specific IFN-γ secreting cell counts between the rKVAC85B prime-boost and the BCG-rKVAC85B boost groups. Although the experiments were conducted independently, they were performed using identical cell counts, enabling us to make direct comparisons based on these counts. Our analysis revealed that the BCG-rKVAC85B vaccination group showed significantly higher IFN-γ secreting cell counts in both the spleen and lungs compared to other groups, demonstrating the efficacy of the BCG-rKVAC85B boosting strategy.

Additionally, we conducted analyses of IgG ELISA OD values, finding that the mean OD value for the rKVAC85B group was approximately 0.875 (Fig 2B), whereas the BCG-rKVAC85B group exhibited a mean OD value of about 1.380 (Fig 3B). This indicates that the IgG responses in the BCG-rKVAC85B group were approximately 1.58 times higher than those observed in the rKVAC85B group. Moreover, the mean IFN-γ ELISPOT responses were about 437.33 spots per 10^6 cells for the rKVAC85B group and approximately 1688 spots per 10^6 cells for the BCG-rKVAC85B group, demonstrating that the BCG-rKVAC85B group exhibited IFN-γ responses approximately 3.86 times greater than those elicited by the rKVAC85B single vaccination (Fig S1). These findings underscore that the BCG-rKVAC85B vaccination not only significantly enhances IFN-γ responses but also boosts IgG antibody production, corroborating the effectiveness and necessity of BCG boosting in the immune modulation against tuberculosis.

Thank you once again for your valuable feedback. We look forward to your response and hope that our revisions meet your expectations.

2. Also the lack of data on the basal levels of the various immunological parameters analyzed as well as the disparity of the vaccination protocols used do not allow to clarify what was actually modulated.

Reply: Thank you for highlighting the gaps in our dataset concerning the basal levels of immunological parameters and the variations in vaccination protocols. We acknowledge the importance of these baseline measurements, particularly in the poly-functional T cell response analysis, which unfortunately was not included in our initial study design.

The constraints of animal model experimentation, particularly under the stringent ethical guidelines that govern tuberculosis research, prevented us from conducting repeat experiments to address this oversight. Nonetheless, we have strived to provide a robust comparison by benchmarking against the existing BCG vaccine, which serves as the current standard in tuberculosis immunotherapy.

We recognize the critical nature of your feedback and will ensure more thorough baseline data integration in future experimental protocols to deepen the impact and clarity of our findings.

3. Furthermore, the histology figures of the lungs of the infected mice do not seem very consistent with what has been reported in panel D of Figure 7.

Reply: Thank you for your feedback regarding the histology figures in our manuscript, specifically the inconsistencies noted in panel D of Figure 7. We recognize that the limited presentation of histopathological data may have restricted our interpretative scope. Our findings indicate smaller lesion sizes predominantly in the BCG-rKVAC85B group, compared to larger lesions in the BCG and PBS groups. This suggests a variation in lesion severity aligned with the inflammation analysis in Figure 7D, though it is not directly conclusive. To provide clearer evidence, we included additional data in Figure 7C, showcasing more pronounced differences among the groups to support our conclusions.

---

## [Editor Report · Decision Letter 2]

17 Mar 2025

Efficacy and immunogenicity of rKVAC85B in a BCG prime-boost regimen against H37Rv and HN878 Mycobacterium tuberculosis strains

PONE-D-24-18487R2

Dear Dr. Jeong,

We’re pleased to inform you that your manuscript has been judged scientifically suitable for publication and will be formally accepted for publication once it meets all outstanding technical requirements.

Kind regards,

Syed Faisal, PhD

Academic Editor

PLOS ONE
---

## [Editor Report · Acceptance letter]

PONE-D-24-18487R2

PLOS ONE

Dear Dr. Jeong,

I'm pleased to inform you that your manuscript has been deemed suitable for publication in PLOS ONE. Congratulations! Your manuscript is now being handed over to our production team.

Kind regards,

on behalf of

Dr. Syed Faisal

Academic Editor

PLOS ONE